# Accurate mapping of mitochondrial DNA deletions and duplications using deep sequencing

**Swaraj Basu**[1‡], **Xie Xie**[1‡], **Jay P. Uhler**[1], **Carola Hedberg-Oldfors**[2], **Dusanka Milenkovic**[3], **Olivier R. Baris**[4,5], **Sammy Kimoloi**[4], **Stanka Matic**[3], **James B. Stewart**[3,6], **Nils-Göran Larsson**[7], **Rudolf J. Wiesner**[4,8], **Anders Oldfors**[2], **Claes M. Gustafsson**[1], **Maria Falkenberg**[1‡]*, **Erik Larsson**[1‡]*

1 Department of Medical Biochemistry and Cell Biology, Institute of Biomedicine, The Sahlgrenska Academy, University of Gothenburg, Gothenburg, Sweden, 2 Department of Laboratory Medicine, Institute of Biomedicine, The Sahlgrenska Academy, University of Gothenburg, Gothenburg, Sweden, 3 Max Planck Institute for Biology of Ageing, Cologne, Germany, 4 Center for Physiology and Pathophysiology, Institute of Vegetative Physiology, University of Cologne, Cologne, Germany, 5 Equipe MitoLab, UMR CNRS 6015, INSERM U1083, Institut MitoVasc, SFR ICAT 4208, Université d'Angers, Angers, France, 6 Wellcome Centre for Mitochondrial Research, Newcastle University Biosciences Institute, Faculty of Medical Sciences, Newcastle University, Newcastle Upon Tyne, United Kingdom, 7 Division of Molecular Metabolism, Department of Medical Biochemistry and Biophysics (MBB), Karolinska Institute, Stockholm, Sweden, 8 Cologne Excellence Cluster on Cellular Stress Responses in Aging-associated Diseases (CECAD), University of Cologne, Cologne, Germany

‡ SB and XX share first authorship on this work. MF and EL are joint senior authors on this work.
* maria.falkenberg@medkem.gu.se (MF); erik.larsson@gu.se (EL)

## Abstract

Deletions and duplications in mitochondrial DNA (mtDNA) cause mitochondrial disease and accumulate in conditions such as cancer and age-related disorders, but validated high-throughput methodology that can readily detect and discriminate between these two types of events is lacking. Here we establish a computational method, MitoSAlt, for accurate identification, quantification and visualization of mtDNA deletions and duplications from genomic sequencing data. Our method was tested on simulated sequencing reads and human patient samples with single deletions and duplications to verify its accuracy. Application to mouse models of mtDNA maintenance disease demonstrated the ability to detect deletions and duplications even at low levels of heteroplasmy.

## Author summary

Deletions in the mitochondrial genome cause a wide variety of rare disorders, but are also linked to more common conditions such as neurodegeneration, diabetes type 2, and the normal ageing process. There is also a growing awareness that mtDNA duplications, which are also relevant for human disease, may be more common than previously thought. Despite their clinical importance, our current knowledge about the abundance, characteristics and diversity of mtDNA deletions and duplications is fragmented, and based to large extent on a limited view provided by traditional low-throughput analyses.

**Data Availability Statement:** The mouse sequencing data has been deposited in the European Nucleotide Archive (ENA) under accession PRJEB37552. The patient sequencing

data has been deposited in the European Genome-phenome Archive (EGA) under accession EGAS00001004380.

**Funding:** The work described here was supported by the Swedish Research Council (2018-02439 to M.F., 2017-01257 to C.M.G., and 2018-02852 to E. L.), the Swedish Cancer Foundation (2019-816 to M.F., 2017-631 to C.M.G., and 2018-747 to E.L.), the Knut and Alice Wallenberg Foundation (KAW 2017.0080 to M.F. and KAW 2015.0144 to E.L.), the European Research Council (683191 to M.F.) and grants from the Swedish state under the agreement between the Swedish government and the county councils, the ALF agreement (ALFGBG-727491 to M.F., and ALFGBG-728151 to C.M.G). The funders had no role in study design, data collection and analysis, decision to publish, or preparation of the manuscript.

**Competing interests:** The authors have declared that no competing interests exist.

Here, we describe a bioinformatics method, MitoSAlt, that can accurately map and classify mtDNA deletions and duplications using high-throughput sequencing. Application of this methodology to mouse models of mitochondrial deficiencies revealed a large number of duplications, suggesting that these may previously have been underestimated.

## Introduction

Mitochondria contain a separate genome which encodes essential subunits of the oxidative phosphorylation system and the RNA molecules (ribosomal and transfer RNA) needed for mitochondrial translation. Mitochondrial DNA (mtDNA) in humans is a small 16.6 kb circular molecule with only a few non-coding regions[1,2]. Thus, large deletions and duplications in mtDNA almost invariably lead to disruption of mitochondrial gene function. These types of structural alterations can be spontaneous or attributed to mutations affecting the nuclear-encoded mtDNA maintenance machinery, e.g. the mitochondrial DNA polymerase γ (POLγ) [3] or the replicative Twinkle helicase[4,5]. Deletions are a common cause of mitochondrial disorders[6–9] while also being linked to cancer[10–12], diabetes[13,14], neurodegenerative disorders[15,16], and the ageing process[16,17]. Duplications are less commonly described, but have for instance been described in patients with disease-causing mutations in *MGME1* [18,19] or in mice expressing a proof-reading-deficient version of Polγ[20].

Despite the clinical significance of mtDNA structural alterations, our current knowledge about their abundance, diversity and exact localization is fragmented. A significant challenge is the multi-copy nature of mtDNA, with each cell containing hundreds to thousands of individual molecules. Most mtDNA alterations are heteroplasmic, meaning that wild-type mtDNA co-exists with mutant variants[21]. This complex DNA landscape makes the molecular characterization of mtDNA variants difficult, with low-level heteroplasmic variants being particularly hard to detect. The most commonly used detection methods, Southern blotting and long-range PCR[22], have limited resolution and cannot define all mtDNA variants in a given sample[23]. Even a variant present at high levels can remain undetected depending on the selection of primers, probes or restriction enzymes, and in the past, using these methods, duplications have wrongly been classified as deletions[18,19,24,25].

An attractive idea is therefore to use high-throughput sequencing to detect mtDNA deletions and duplications, as this potentially can provide more sensitive, less biased and more accurate mapping of these alterations. This would also dramatically simplify the workflow, and would enable exploration of mtDNA deletions and duplications in a large body of preexisting sequencing datasets. Due to the high copy number of mtDNA in cells ($n$ = 1,000–10,000), mtDNA-derived reads are typically highly abundant in genomic sequencing data, in principle making the technology ideally suited for the purpose. The basic bioinformatics principles for determining structural alterations from short read sequencing are well-known, specifically identification of discordant paired-end reads or gapped/split alignment of individual reads to the reference genome. However, details in the implementation may have a large influence on performance, and tools for mapping structural changes in the nucleus show a surprising degree of discordance[26]. While the small size of the mitochondrial genome simplifies the problem, it is made harder by the fact that mitochondrial deletions commonly occur near repetitive sequences, and mapping of structural events on a circular genome presents additional challenges.

Several methods have recently been developed specifically for identification of mtDNA deletions from high-throughput short read sequencing, including MitoDel[27], Splice-Break

[28], eKLIPse[29], MitoMut[30], and a PERL script provided in (Zambelli et al., 2017)[31]. These methods rely on gapped alignments to predict deletions, but fail to recognize that every such event can represent either a deletion or a duplication affecting the arc complementary to the deleted part; a consequence of the circularity of mtDNA. Duplications can form as a consequence of mutations in mitochondrial replication factors, and correct identification and classification of such alterations is therefore an important requirement for any bioinformatics method pertaining to analysis of mtDNA structural changes.

Here we present the first high-throughput computational pipeline, MitoSAlt (Mitochondrial Structural Alterations), for identification, quantification and visualization of both deletions and duplications in mtDNA. The performance of MitoSAlt was carefully established using simulated sequencing data, patient samples with single events, and mouse models of mtDNA maintenance disease. MitoSAlt also introduces a way of visualizing the results such that duplications and deletions, as well as start and end positions, are unambiguously indicated. Using MitoSAlt, we also demonstrate that disease-causing mutations affecting specific steps in mtDNA replication cause distinct structural alterations in mtDNA.

## Results

### Detection of deletions and duplications with MitoSAlt

MitoSAlt is designed to take single- or paired-end sequencing reads as input to generate a map of predicted deletions and duplications, visualized in a circular plot along with tab delimited tables detailing the breakpoint positions and heteroplasmy levels (**Fig 1A**, further detailed in Materials and Methods). The pipeline relies on an initial alignment of sequencing reads to the nuclear and mitochondrial (Mt) genome using HISAT2[32] to remove nuclear reads while retaining mtDNA-mapped and unmapped reads. This step accelerates the analysis, but may optionally be disabled when working with species having extensive nuclear mitochondrial DNA (NUMT) regions such as mouse[33] to avoid patch-wise reduced mtDNA read coverage. This is followed by alignment to mtDNA using LAST[34], processing of the LAST results to identify deletions and duplications based on split alignments, and classification of deletions and duplications along with plotting the results and generating final tables. Additionally, in the case of whole genome sequencing (WGS), when no mtDNA or nuclear enrichment has been

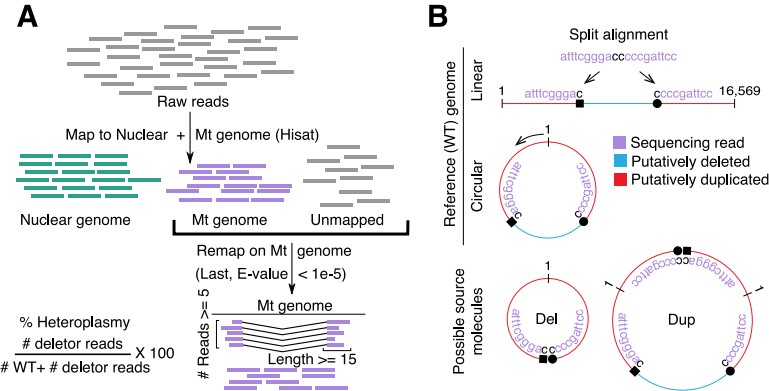

**Fig 1. MitoSAlt pipeline overview.** (**A**) Raw sequencing reads are mapped first to the nuclear and mitochondrial (Mt) genomes using a fast aligner, followed by precision alignment of unmapped and Mt mapped reads to the Mt genome to identify "split" reads informative of structural breakpoints. (**B**) Dual interpretation of split alignments: a split read can represent either a deletion or a complementary arc duplication, and these scenarios are indistinguishable using short-read sequencing.

performed, MitoSAlt can compare mitochondrial and nuclear read counts to estimate relative mtDNA levels, which are indicative of mtDNA copy number.

Similar to other methods, MitoSAlt relies on identification of reads aligning in a split/gapped fashion to the linear mitochondrial genome (**Fig 1A**). However, it is important to note that on a circular genome, every split read can represent either a deletion or, alternatively, a duplication of the mtDNA arc complementary to the deletion, and these two possibilities are indistinguishable when using short read sequencing (**Fig 1B**). MitoSAlt handles this by initially assuming that all events are deletions, followed by complementation and re-classification as a duplication in cases where the altered mtDNA molecule is deemed incapable of replicating due to loss of one or both origins (OriH or OriL; positions are user-definable). The favored interpretation is thus one where both origins are unaltered or, when this is not possible, none are deleted. The deletion/duplication classification is always non-ambiguous, since only one interpretation will satisfy these criteria while the other will violate them. Furthermore, the circularity of mtDNA implies that both deletions and duplications can produce alignments where the split segments map in reverse order to the linear reference, and care has been taken for MitoSAlt to handle and interpret this correctly (**S1 Fig**).

## Evaluation of MitoSAlt on simulated sequencing data

We first evaluated the ability of the pipeline to accurately detect and classify both duplications and deletions based on a small set of simulated alterations present at high heteroplasmy levels. These were designed to cover the main classes of conceivable events that may still maintain mtDNA replicability. To this end, deletions (2,001–3,999 and the so-called common deletion [35] at 8,470–13,446; coordinates indicate start and end of the affected segment) and duplications (16,069–500, 2,500–3,500, 5,000–6,000, and 9,000–10,000) were introduced into the human reference mitochondrial genome (rCRS), each one at 16.7% heteroplasmy. These were combined with the nuclear genome to emulate a mitochondrial copy number of 6,000, and 10 million reads (5 million 2 × 126 bp) were generated using a model that emulates Illumina HiSeq characteristics[36]. Both alignment steps were performed (nuclear and mtDNA using HISAT2 followed by LAST on unmapped and Mt aligned reads). Eventually, 98.4% of mitochondrial reads ($n$ = 210,235) were mapped to mtDNA, resulting in a mean coverage of ~1,600× (**Fig 2A**).

MitoSAlt accurately detected all events at single bp resolution and correctly classified them as deletions or duplications, with heteroplasmy estimates varying between 12.5% and 16.7% (**Fig 2A**). Between 148 and 213 reads were correctly aligned across each breakpoint (theoretical expectation 266 without any dropouts), while a smaller number of alignments (0–4 reads) supported breakpoints within 5 bp of the actual positions (**Fig 2A**). No other events were detected despite inclusion of nuclear chromosomes in the simulations. These results support that MitoSAlt can accurately identify and classify deletions and duplications without additional false positive detections.

We further compared the performance of MitoSAlt with five published pipelines on the same simulated dataset (**S1 Table**). Two of the tools, eKLIPse and the PERL script provided in (Zambelli et al., 2017), identified all events, but with the duplications reported as complementary arc deletions (i.e. start and end coordinates in reverse order). eKLIPse and Zambelli et al identified at most 94 and 186 breakpoint-spanning reads, respectively, suggesting that eKLIPse in particular has reduced sensitivity compared to MitoSAlt. Zambelli et al was less accurate when breakpoints were flanked by repeats: the 8,470–13,446 common deletion (flanked by a 13 bp identical repeat) was reported at 8482–13,447, and the 2,500–3,500 duplication (flanked by a longer imperfect repeat) was reported as a deletion at 3,525–2,500. The remaining tools

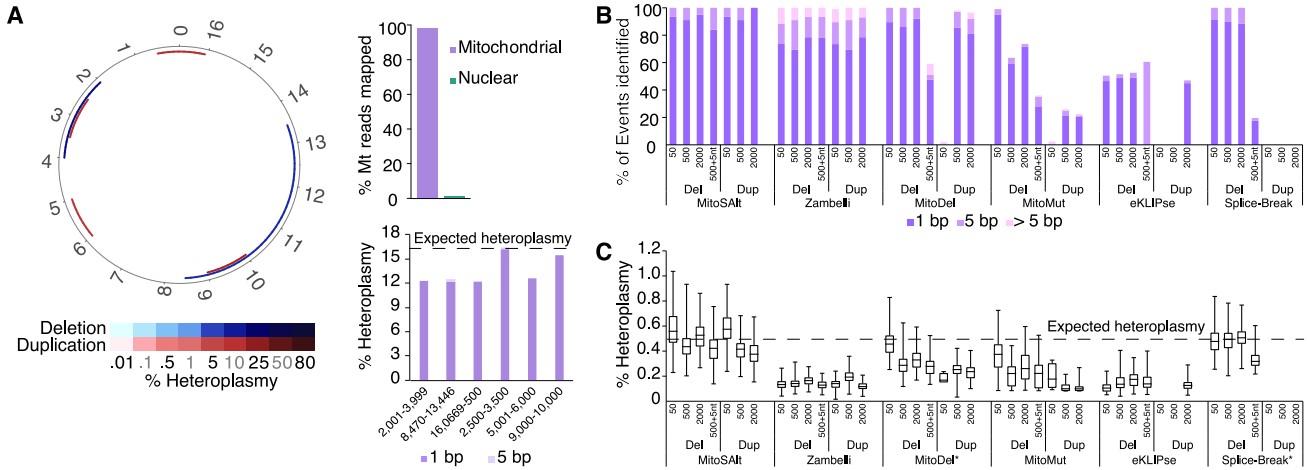

**Fig 2. MitoSAlt pipeline performance on simulated data.** (**A**) Evaluation on simulated sequencing data harboring two synthetic deletions and 4 duplications, each at 16.7% heteroplasmy (2 × 126 bp, 10,000,000 reads, resulting in ~2,000× mtDNA coverage). The circular plot shows deleted (blue) or duplicated (red) segments. The upper bar graph indicates the fraction of Mt reads mapped to the mitochondrial genome, while the lower shows heteroplasmy levels estimated by MitoSAlt for each event (events with 1 bp or 5 bp of the expected breakpoints are quantified separately). (**B**) Evaluation of sensitivity on simulated sequencing datasets containing large numbers of low heteroplasmy deletions and duplications of various sizes. Each data set contained 200 minor or major arc events, each at 0.5% heteroplasmy (2 × 126 bp, 50,000,000 reads, resulting in ~6,000× mtDNA coverage). "500+5nt" refers to 500 bp deletions with 5 bp non-template random insertions at the breakpoint. (**C**) Box and whisker plot of heteroplasmy levels estimated by different pipelines. The boxes show 25th to 75th percentiles, and whiskers show the minimum and maximum value. *, These tools do not directly report heteroplasmy levels, and estimates were instead made based on the reported number of reads supporting each event and the average mitochondrial read-depth.

identified 4 out of 6 events at best. Splice-Break specifically failed to identify duplications associated with inverse-order split alignments (**S1 Fig**), suggesting that the algorithm is not designed to handle this case. In addition to finding 4 out of the 6 true alterations, MitoMut identified 4 additional small deletions (**S1 Table**). Similar results were obtained when simulated reads were generated using an error model derived from empirical data (**S1 Table**).

Next, we generated simulated datasets containing large numbers of low heteroplasmy level (0.5%) deletions or duplications of various sizes (50, 500 and 2000 bp). Each dataset contained 200 events of a single type distributed across the major and minor arcs. Additionally, a dataset with 500 bp deletions with 5 bp random insertions was generated, to test the ability to handle non-template insertions at breakpoints. Mitochondrial number was set to 6,000, and 50 million reads (5 million 2 × 126 bp) were generated for each dataset, resulting in a mean coverage of ~5,900× on chrM. All events were detected by MitoSAlt and Zambelli et al, though the latter had lower accuracy with respect to exact determination of breakpoint coordinates (**Fig 2B** and **S1 Table**). Remaining tools all showed reduced or no sensitivity with respect to small duplications or duplications in general, as well as deletions with non-template insertions. Heteroplasmy estimates reported by MitoSAlt ranged from 0.38% to 0.57% on average in each dataset (**Fig 2C**). No events were detected by MitoSAlt or the other tools in a simulated wild type dataset of similar size. MitoSAlt thus compared favorably to other tools in terms of sensitivity and breakpoint coordinate accuracy, in addition to being the only method capable of differentiating between duplications and deletions.

## Application to mitochondrial disease patients

We next tested the MitoSAlt pipeline on muscle biopsy DNA from mitochondrial disease patients with single high-heteroplasmy mtDNA deletions or duplications present at high levels as detected by long-range PCR (LX-PCR). Two patients carried a deletion while the third

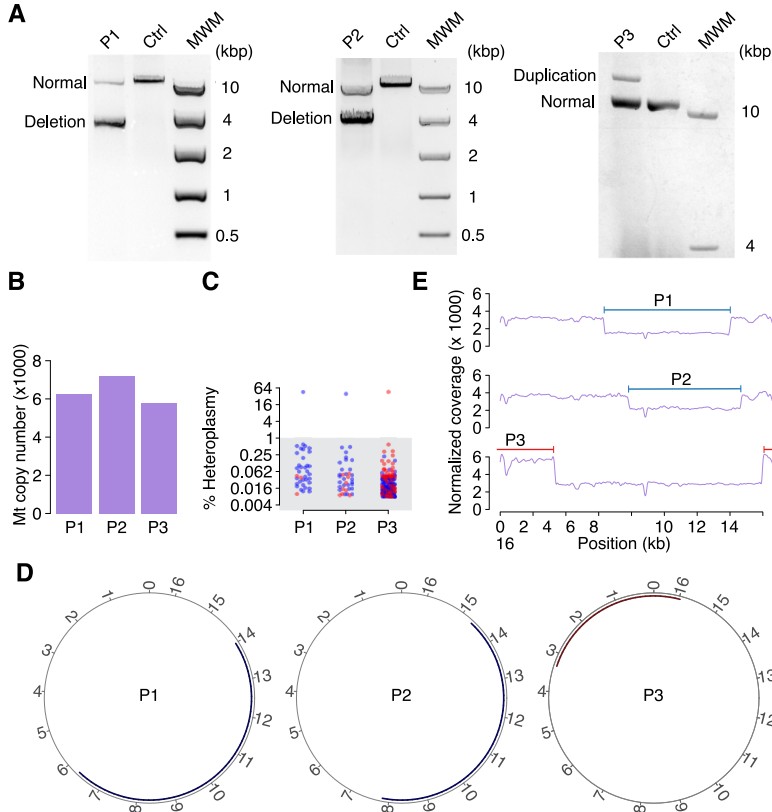

**Fig 3. Assessment of MitoSAlt on patient samples with a single deletion or duplication.** (**A**) Total DNA from patients (P1, P2 and P3) and controls were analyzed by LX-PCR using two different primer sets. A single deletion was detected in patients P1 and P2 using primers LX1 and LX2, while a single duplication was detected in P3 with primers LX3 and LX4. Amplicons from wild type mtDNA (denoted "normal") were also detected in all patients. (**B**) Predicted mtDNA copy number in the patients. (**C**) Heteroplasmy levels for the identified deletions/duplications (marked in blue and red, respectively) in the patient samples. All cases have single events at heteroplasmy levels (> 35%), in addition to multiple low-heteroplasmy alterations (<1%, grey area). (**D**) Circular plots showing deletions/duplications at heteroplasmy >1%, all being consistent with the LX-PCR results. (**E**) Read coverage depth across the Mt genome for the human samples shows drastic changes in the regions identified as being deleted or duplicated (marked in blue and red respectively). MWM, molecular weight marker.

patient had a duplication (**Fig 3A**). WGS resulted in a coverage between 83,737× and 121,703× on chrM, and the estimated mtDNA levels, which can be used to predict mtDNA copy number, varied between 5,789 and 7,204 for all samples (**Fig 3B**). MitoSAlt detected a single high-level heteroplasmy (>50%) deletion or duplication in each patient as expected (**Fig 3C**). Additional low-level heteroplasmy (<1%) events often had breakpoints close to the main alterations, which may represent inaccurate alignments caused by sequencing errors (**Fig 3C**). The major breakpoints predicted by MitoSAlt (deletions at 6,330–13,993, 7,826–14,673 and a duplication spanning the D-loop at 15,973–3,326) were compatible with the LX-PCR results and corresponded closely to breakpoints estimated from chrM read depth (**Fig 3D and 3E**).

Additionally, we tested the MitoSAlt pipeline on whole genome sequencing data from three human tumors (deriving from liver, pancreas and skin), where read depth-based analysis previously suggested presence of large mtDNA duplications or deletions [37], and found that these events were confirmed by our approach (**S2 Fig**). These results provide further support that MitoSAlt can correctly identify breakpoints and classify events as deletions or duplications based on retention or loss of replication origins.

## MitoSAlt detects large numbers of duplications in mouse models of mtDNA disease

Having validated the MitoSAlt pipeline on patients carrying single large-scale mtDNA duplications or deletions, we decided to extend our analysis to more complex DNA samples. To this end, we obtained DNA from mice previously shown to harbor multiple mtDNA structural alterations due to mutations in the gene for the Twinkle helicase (*Twnk*[K320E]; two different mice, M1 and M2), knockout of the mtDNA maintenance exonuclease Mgme1 (*Mgme1*[-/-]; two different mice, M3 and M4), or mutations in the exonuclease domain of DNA polymerase gamma Polγ (*Polg*[D257A]; one mouse, M5). All three genes are important for mtDNA maintenance in mice and in humans[19,24,38–40]. The mutant mouse samples (M1-M5) and wild-type controls (C1-C5) were subjected to WGS (*Twnk*[K320E] and *Mgme1*[-/-]) or sequencing following an mtDNA enrichment protocol (*Polg*[D257A]), resulting in a coverage on chrM ranging from 35,913× to 150,182× (**S3 Fig**).

MtDNA level estimates for the *Twnk*[K320E] and *Mgme1*[-/-] mutants were comparable to wild type control samples (**Fig 4A** and **S1 Table**), while the use of mtDNA enrichment precluded mtDNA level estimation in the *Polg*[D257A] mutant sample. A large number of events were detected in all mutant samples (ranging from 95 to 4841), mostly duplications present at low

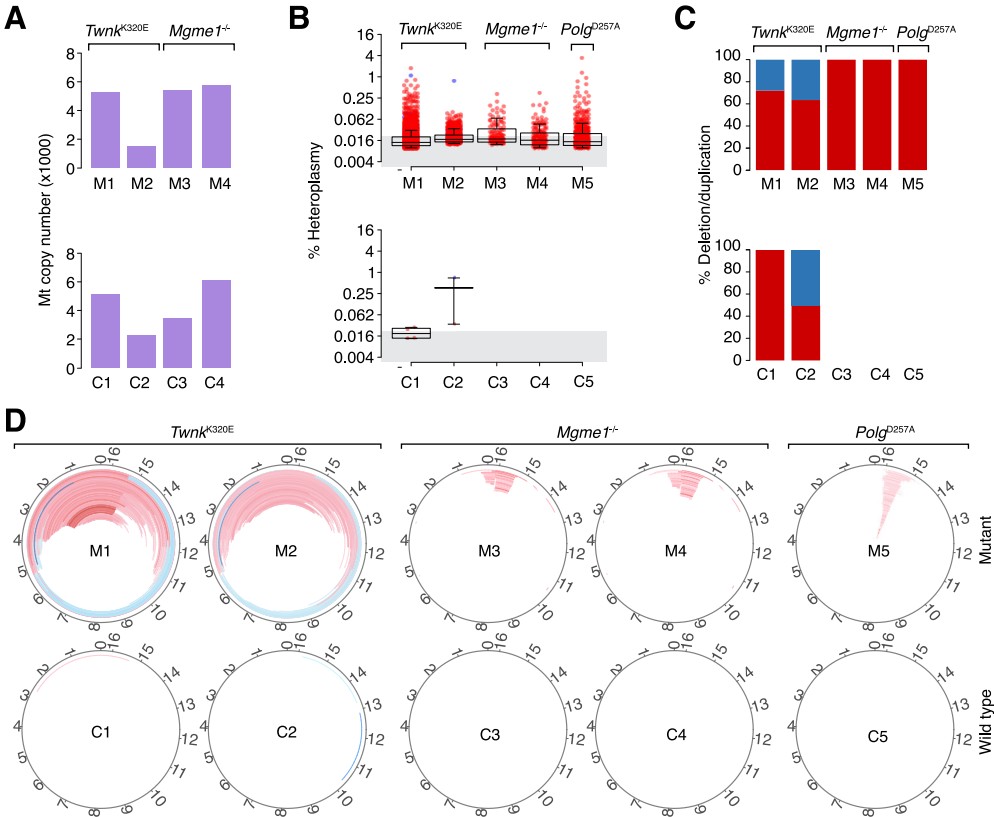

**Fig 4. Identification of mtDNA structural alterations in wild-type and *Mgme1*, *Twnk* or *Polg* mutant mice using the MitoSAlt pipeline.** (**A**) Predicted copy number for the given mutant and wild-type samples (denoted M and C, respectively). Copy number could not be estimated for the *Polg* samples due to use of an mtDNA enrichment protocol. (**B**) Heteroplasmy levels for the deletions (blue) and duplications (red) identified in the mutant and wild-type samples. The grey area delineates low-heteroplasmy events (<0.02%). (**C**) Fraction deletions (blue) and duplications (red) in each sample. (**D**) Circular plots showing the deletions (blue) and duplications (red) identified in the mutant and wild-type samples. For visual clarity, a heteroplasmy cut-off of 0.02% was used for all samples.

heteroplasmy levels (maximum 3.47% and with the average per sample ranging from 0.023% to 0.038%; **Fig 4B and 4C**). In contrast, the negative control samples were essentially void of structural events (in total 5 events, all below 0.01%; **Fig 4B and 4C**).

Visualization of the events on the circular Mt genome revealed two distinct patterns, where *Mgme1*[-/-] and *Polg*[D257A] shared a common signature involving multiple, shorter duplications in the non-coding region (NCR), while *Twnk*[K320E] instead was characterized by abundant longer duplications, spanning from a hotspot in the NCR to another hotspot in the middle of the minor arc (**Fig 4D**). These alteration signatures may reflect similarities and differences in the underlying molecular processes leading to breakpoint formation.

## Discussion

MitoSAlt is the first pipeline explicitly designed to identify and correctly classify both deletions and duplications in mtDNA. MitoSAlt also provides a novel way of visualizing complex mtDNA alteration patterns, where deletions and duplications are unambiguously indicated along with their start and end positions and heteroplasmy levels. While primarily designed to be used on genomic sequencing data (whole genome or mtDNA enriched), MitoSAlt may in principle also be applicable to transcriptome or exome sequencing data, although the latter often exhibits limited mtDNA coverage. The performance of MitoSAlt was verified using simulated sequencing data, which showed that low heteroplasmy (0.5%) events are detectable with high sensitivity even at moderate sequencing depths. MitoSAlt was further applied to sequence data from human patients carrying single deletion/duplication events confirmed by LX-PCR, and mutant mice strains previously shown to harbor large numbers of mtDNA structural alterations[18,20,24,38].

Results from LX-PCR analysis of mice expressing *Twnk*[K320E] (corresponding to the disease causing mutation *TWNK*[K319E] in humans) have previously been interpreted as evidence for mtDNA deletions [39]. Interestingly, MitoSAlt instead predicted far more duplications (more than 85%) than deletions (less than 15%) in *Twnk*[K320E] mice. Duplications also outnumbered deletions in mice with full-body knockout of *Mgme1*, recapitulating patients with homozygous nonsense mutations in *MGME1*[18], and in mice expressing exonuclease deficient Pol γ, *Polg*[D257A], which were previously proposed to harbor duplications in the same region [20]. Our results thus support that mtDNA duplications may be prevalent.

The mechanisms underlying mtDNA deletion formation have been carefully studied, leading to different models, including copy-choice recombination[41] and double-strand break repair[42]. How duplications are formed, and which enzymes are responsible, is still unclear, but the detailed data provided by MitoSAlt can be a useful resource for developing mechanistic hypotheses. For example, the similar alteration patterns seen in *Mgme1*[-/-] and *Polg*[D257A] mice (short duplications in the NCR) could indicate that these two enzymes are required for a common molecular function, a conclusion supported by previous studies, which have linked Mgme1 and Polγ to the formation of ligatable nicks during termination of mtDNA replication in the NCR[43–45].

MitoSAlt also estimates relative mtDNA levels, which are indicative of mtDNA copy number. However, for a more accurate mtDNA copy determination, the presence of large structural alterations in mtDNA must be considered. For example, long deletions present at high heteroplasmy will lead to a drop in mtDNA levels, even if the mtDNA copy number remains unchanged. In a related way, mtDNA copy number drops in the *Mgme1*[-/-] mice [18], but mtDNA levels remain unchanged due to the constant production of long, linear mtDNA fragments that cannot be replicated or expressed.

In conclusion, MitoSAlt is carefully validated tool for precision mapping of mtDNA structural alterations, specifically designed to detect and discriminate between deletions and

duplications. MitoSAlt will facilitate further dissection of the mechanistic basis underlying the formation of these types of events, and will enable detailed analysis of samples from patients with mitochondrial diseases.

# Materials and methods

## Ethics statement

The transgenic mice studies were approved by the Landesamt für Natur, Umwelt und Verbraucherschutz Nordrhein–Westfalen (reference numbers 84–02.04.2015.A103, 84–02.05.50.15.004 and 2013-A165) and performed in accordance with the recommendations and guidelines of the Federation of European Laboratory Animal Science Associations (FELASA). Human patients gave informed consent for the investigations made and the study was approved by the Regional Ethics Committee at the University of Gothenburg, Sweden (number 390–07).

## MitoSAlt

The MitoSAlt pipeline is comprised of three modules combined into a single pipeline: (1) alignment of sequencing reads (using PERL wrapper third party softwares), (2) parsing aligned reads to identify Mt breakpoints (PERL and R), and (3) plotting the results on the circular Mt genome and analysis of breakpoint repeats (R programming environment).

## Alignment of sequencing reads

The raw sequencing reads are aligned to the source genome (Nuclear + Mitochondrial) using HISAT2[46]. HISAT2 is run with default parameters for RNA sequencing and specific parameters are used to customize it for DNA sequencing (—no-temp-splicesite—no-spliced-alignment—max-intronlen 5000). Following the first round of alignment the reads which remain unmapped or are mapped to the mitochondrial genome are extracted and converted to a concatenated FASTQ using Samtools. The FASTQ is realigned to the mitochondrial genome using the lastal (-Q1 -e80), processed using last-split and converted from MAF to BAM and TAB format using maf-convert, where all the binaries are part of the LAST software package. The results in TAB format are parsed in PERL and R to classify the potential deletions and duplications. If the input sequencing data is enriched for mitochondrial DNA/RNA, then the pipeline skips the initial HISAT2 mapping and concatenates the FASTQ files using reformat. sh from BBMap software suite and maps the concatenated reads on the mitochondrial genome using LAST, where the downstream processing remains the same.

## Parsing aligned reads to identify Mt breakpoints

The TAB formatted output is parsed in PERL to remove duplicated reads (both wildtype and mutant) and generate three output files a) BED format file with the list of split reads which may support a deletion or a duplication b) BREAKPOINT file with the list of breakpoints identified c) CLUSTER file, which groups the breakpoints at a given distance threshold and estimates the heteroplasmy at a given pair of clustered breakpoints as the ratio of reads supporting the breakpoints by the number of wildtype reads overlapping the breakpoints.

## Final report and circular plots

The CLUSTER, and BREAKPOINT files are further used by an R script to generate a final table, classifying each cluster as a duplication or a deletion using the logic described in **S1 Fig**. This report also contains information about direct repeat sequences overlapping with or

flanking the breakpoints. It should be noted that genomic coordinates in the final table refer to start and end positions of the deleted or duplicated segments, rather than junction coordinates. Finally, the breakpoint positions (at the cluster level) are plotted on a circular plot (size of the input mitochondrial genome) as arcs using the R plotrix package, where the individual arcs are colored to indicate whether they represent deletions or duplications, and where the estimated heteroplasmy is indicated by the intensity of the color.

### Generation of simulated sequencing data

For the initial evaluation, involving a limited number of high heteroplasmy level events, six mutant mitochondrial reference genomes were generated, each containing a large deletion or duplication as detailed in Results. These were concatenated such that each would be present at a heteroplasmy of 16.7% and included in multiple copies together with the nuclear human chromsomes (hg19 assembly) to emulate an mtDNA copy number of 6000. Next we generated simulated reads using *InSilicoSeq*, a Python software package[36]. Two different error models were used: the default Illumina Hiseq model (10,000,000 2×126 bp paired-end reads) and an empirical error model base on NextSeq 500 generated whole genome sequencing data (6,000,000 2×76 bp paired-end reads). To evaluate the performance on a larger number of low heteroplasmy events, 6 separate datasets were generated, each containing 200 events as described in Results. These datasets were generated by concatenating mitochondrial genomes containing different deletions or duplications such that each would have a heteroplasmy level of 0.5%. These were combined with a nuclear human genome to emulate mtDNA copy number of 6,000. Simulated reads were generated using the Illumina HiSeq Model (50,000,000 2×126 bp paired-end reads).

### DNA samples

For the LX-PCR and MitoSAlt analyses of human samples, total DNA was isolated from muscle biopsies from three patients with mitochondrial disease (Patient 1; age 9, Patient 2; age 16 and Patient 3; age 58) and age-matched control individuals using standard protocols. For MitoSAlt analysis of murine samples the following mice variants were used: *Twkn*^K320E transgenic mice expressing a dominant-negative mutant version of the *Twinkle* gene in skeletal muscle, which were generated by crossing R26-K320E-Twinkle^loxP/+ mice[39] with Mlc1f-cre mice[47], *PolgA*^D257A mice carry a point mutation in the 3'-5' exonuclease domain of the replicative DNA polymerase POLG[38], and *Mgme1*^-/- knockout mice are deficient in the MGME1 exonuclease[24]. Total DNA was isolated from muscle for *Twkn*^K320E analysis, from heart for Mgme1-/- analysis, and mtDNA was isolated from heart for PolgAD257A analysis using standard techniques.

### LX-PCR

LX-PCR was performed on total DNA extracted from human muscle specimens to detect possible large scale mtDNA deletions and/or duplications using GoTaq Long PCR Master Mix according to the manufacturer's protocols (Promega, Madison WI, USA). The mtDNA was amplified with two sets of primers: set 1, LX1_m.5420-5447 (TGA ACA TAC AAA ACC CAC CCC ATT CCT C) and LX2_m.16232-16259 (GTG GCT TTG GAG TTG CAG TTG ATG TGT G) and set 2, LX3_m.8020-8000 (CGG GAG TAC TAC TCG ATT GTC) and LX4_m.13940-13972 (GCA CAA TCC CCT ATC TAG GCC TTC TTA CGA GCC) resulting in PCR products of size 10.8 kb and 10.6 kb, respectively, based on wild type mtDNA. PCR products were analysed by electrophoresis on 0.6% agarose gels.

### Illumina sequencing

The patient samples were sequenced at Science for Life Laboratory in Stockholm, Sweden, using an Illumina NovaSeq 6000, resulting in 647.8–901.2 million 2×150 bp reads. The *Polg* mouse samples were sequenced at the Max Planck Genome Center in Cologne, Germany, using an Illumina HiSeq 2500, resulting in 100.7–101.4 million 2x250 bp reads. The *Twnk* and *Mgme1* mouse samples were sequenced at the Genomics Core Facility at the Sahlgrenska Academy in Gothenburg, Sweden, using an Illumina NovaSeq 6000, resulting in 602.7–691.5 million 2x150 bp reads.

### Software availability

MitoSAlt is available through SourceForge at https://sourceforge.net/projects/mitosalt.

## Supporting information

**S1 Fig. Both deletions and duplications may give rise to forward or reverse order split alignments on a circular genome.** The circularity of mtDNA presents special challenges when it comes to handling gapped/split alignments of short reads. Both deletions and duplications may give rise to split alignments where the split segments align in both forward or reverse order on the linear genome, depending on the type of alteration and its location relative to position 1, indicated here as $O_H$. Each gapped/split alignment, whether segments are in forward or reverse order, may represent either a deletion or a duplication, and these two possibilities are indistinguishable. Specifically, deletion of a specific segment A or duplication of the segment complementary to A (i.e. the remainder of the circular genome not covered by segment A) will produce identical split read alignments.
(EPS)

**S2 Fig. Circular plots showing deletions/duplications in three cancer genomes.** Estimated heteroplasmies are shown in the center of each circle.
(EPS)

**S3 Fig. Read coverage on mouse chrM for the included samples.**
(EPS)

**S1 Table. Overview of sequenced DNA samples including basic statistics, performance of MitoSAlt compared to five published pipelines based on simulated sequencing data, and additional numerical data underlying figures.**
(XLSX)

## Acknowledgments

We thank Brith Leidvik for technical assistance. We would also like to acknowledge the Clinical Genomics Stockholm facility at Science for Life Laboratory and the Genomics Core Facility at the Sahlgrenska Academy for providing assistance in next generation sequencing. We acknowledge the contributions of the many clinical networks across ICGC and TCGA, enabling the analysis of whole genome sequencing data from human tumors in this study.

## Author Contributions

**Conceptualization:** Claes M. Gustafsson, Maria Falkenberg, Erik Larsson.

**Data curation:** Swaraj Basu, Xie Xie.

**Formal analysis:** Swaraj Basu, Xie Xie.

**Funding acquisition:** Anders Oldfors, Claes M. Gustafsson, Maria Falkenberg, Erik Larsson.

**Investigation:** Jay P. Uhler, Carola Hedberg-Oldfors, Anders Oldfors, Maria Falkenberg.

**Methodology:** Swaraj Basu, Xie Xie, Maria Falkenberg, Erik Larsson.

**Project administration:** Maria Falkenberg, Erik Larsson.

**Resources:** Dusanka Milenkovic, Olivier R. Baris, Sammy Kimoloi, Stanka Matic, James B. Stewart, Rudolf J. Wiesner, Maria Falkenberg.

**Software:** Swaraj Basu, Xie Xie.

**Supervision:** Maria Falkenberg, Erik Larsson.

**Validation:** Nils-Göran Larsson.

**Visualization:** Swaraj Basu, Xie Xie, Jay P. Uhler.

**Writing – original draft:** Swaraj Basu, Xie Xie, Jay P. Uhler.

**Writing – review & editing:** Swaraj Basu, Xie Xie, Jay P. Uhler, Dusanka Milenkovic, James B. Stewart, Nils-Göran Larsson, Rudolf J. Wiesner, Anders Oldfors, Maria Falkenberg, Erik Larsson.

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
