## [Decision Letter · Decision Letter 0]

17 Jun 2020

Dear Dr Larsson,

Thank you very much for submitting your Research Article entitled 'Accurate mapping of mitochondrial DNA deletions and duplications using deep sequencing' to PLOS Genetics. Your manuscript was fully evaluated at the editorial level and by independent peer reviewers. The reviewers appreciated the attention to an important problem, but raised some substantial concerns about the current manuscript. Based on the reviews, we will not be able to accept this version of the manuscript, but we would be willing to review again a much-revised version. We cannot, of course, promise publication at that time.

Should you decide to revise the manuscript for further consideration here, your revisions should address the specific points made by each reviewer, including additional benchmarking of MitoSAlt with respect to sensitivity/specificity, and additional validation as described by both reviewers.. We will also require a detailed list of your responses to the review comments and a description of the changes you have made in the manuscript.

If you decide to revise the manuscript for further consideration at PLOS Genetics, please aim to resubmit within the next 60 days, unless it will take extra time to address the concerns of the reviewers, in which case we would appreciate an expected resubmission date by email to plosgenetics@plos.org.

[LINK]

We are sorry that we cannot be more positive about your manuscript at this stage. Please do not hesitate to contact us if you have any concerns or questions.

Yours sincerely,

Ed Reznik

Guest Editor

PLOS Genetics

Gregory Barsh

Editor-in-Chief

PLOS Genetics

Reviewer's Responses to Questions

**Comments to the Authors:**

Reviewer #1: The authors describe a method for identification of structural variation in mitochondrial genomes. A selling point of the method is its ability to accurately classify SVs as deletions or duplications, in addition to its ability to do so from whole genome, whole exome or transcriptome sequencing data. The authors demonstrate the accuracy of their method on simulated data and the utility when applied to mouse models of mitochondrial disorder.

The paper is clearly and the figure quality is good. I have the following major concerns.

1. As the authors have noted, discerning duplications from deletions is unidentifiable for a circular genome without additional information. Using the two replication origins is reasonable, but depending on the location of the deletion / duplication, there may still be ambiguity. The authors should mention this ambiguity and describe any rules they apply to decide between duplications and deletions in this situation.

2. The simulations as described are insufficient to fully evaluate Mitosalt or competing methods. Unless I misunderstand, the authors simulated dataset included only 6 events. By contrast, MitoMut was benchmarked on a simulated dataset described as follows. “We simulated 3000 paired-end Illumina whole-genome sequencing experiments with one deletion each. Of the simulations, 1000 had small deletions (5-30 bps), 1000 had medium deletions (31-500 bps), and 1000 had large deletions (500-5000 bps).” The authors should benchmark on a more comprehensive dataset, ideally one with similar scale to that described in the MitoMut paper.

3. In the simulation results, no mention is made of the number of false positives produced by mitosalt or the other tools. Mitosalt appears to be more sensitive but how specific is it relative to other methods.

4. The authors have applied their method to WGS and MT enriched WGS sequencing but have not provided any evidence supporting their claim that the method works on whole exome or transcriptome sequencing.

Reviewer #2: In this manuscript entitled "accurate mapping of mitochondrial DNA deletions and duplications using deep sequencing", the authors generated a straightforward tool, or MitoSAlt, to call the mitochondrial structural variations. Structural variations in mitochondrial DNA have not been extensively studied due to technical difficulties. The authors compared their tool with a few publicly available tools, such as MitoDel, Splice-Break, EKLIPse, MitoMut and a Perl script by Zambelli. From the benchmark study, the authors concluded that the performance of MitoSAlt is superior to these tools. I feel that MitoSAlt is very useful and will be used in future mitochondrial genome studies. The manuscript also reads well. With a few additional validations, I think the manuscript is suitable for publication in Plos Genetics.

Minor comments:

(1) Sensitivity: what is the sensitivity of mtDNA structural variation detection of MitoSAlt? I believe that it depends on the mtDNA sequence read-depth and some features of mtDNA structural variants. However, I am still wondering the minimum heteroplasmy of mtDNA variants that can be detected by MitoSAlt in given read depth. Is it able to show any metrics to the authors?

(2) In structural variations, sometimes non-template nucleotide insertions are engaged in the breakpoints. How these sequences are handled in MitoSAlt?

(3) I am wondering how breakpoint sequence microhomology is treated in MitoSAlt calls.

(4) Is there any possibility of false-positives due to hidden NUMTs? For example, if a NUMT sequence is equivalent to a mitochondrial sequence with a large deletion, and the MUMT is not represented in the reference genome, then the sequence will be misaligned to the mitochondrial reference genome and may appear as a mitochondrial DNA structural variation at ~1% heteroplasmic level.

(5) Figure 3. I am wondering whether the authors can further validate the variations identified by MitoSAlt with another technique.

(6) How precise the heteroplasmic level estimates of variant mtDNA?

(7) In a recent paper (Yuan Yuan et al., Nature Genetics 2020, https://www.nature.com/articles/s41588-019-0557-x ), the authors identified mtDNA somatic structural variations in three human cancer genomes. The authors may want to test MitoSAlt to show the performance of their tool.

**Have all data underlying the figures and results presented in the manuscript been provided?**

Reviewer #1: Yes

Reviewer #2: Yes

PLOS authors have the option to publish the peer review history of their article (what does this mean?). If published, this will include your full peer review and any attached files.

Reviewer #1: No

Reviewer #2: Yes: Young Seok Ju at KAIST

---

## [Decision Letter · Decision Letter 1]

2 Nov 2020

Dear Dr Larsson,

We are pleased to inform you that your manuscript entitled "Accurate mapping of mitochondrial DNA deletions and duplications using deep sequencing" has been editorially accepted for publication in PLOS Genetics. Congratulations!

Yours sincerely,

Ed Reznik

Guest Editor

PLOS Genetics

Gregory Barsh

Editor-in-Chief

PLOS Genetics

Comments from the reviewers (if applicable):

Reviewer's Responses to Questions

**Comments to the Authors:**

Reviewer #1: Thank you for addressing my concerns

Reviewer #2: The authors addressed all the queries and the revised manuscript seems to be suitable for publication.

**Have all data underlying the figures and results presented in the manuscript been provided?**

Reviewer #1: Yes

Reviewer #2: Yes

PLOS authors have the option to publish the peer review history of their article (what does this mean?). If published, this will include your full peer review and any attached files.

Reviewer #1: No

Reviewer #2: No

**Data Deposition**

http://datadryad.org/submit?journalID=pgenetics&manu=PGENETICS-D-20-00780R1

**Press Queries**

---

## [Editor Report · Acceptance letter]

30 Nov 2020

PGENETICS-D-20-00780R1 

Accurate mapping of mitochondrial DNA deletions and duplications using deep sequencing 

Dear Dr Larsson, 

We are pleased to inform you that your manuscript entitled "Accurate mapping of mitochondrial DNA deletions and duplications using deep sequencing" has been formally accepted for publication in PLOS Genetics! Your manuscript is now with our production department and you will be notified of the publication date in due course.

With kind regards,

Nicola Davies

PLOS Genetics

On behalf of:
